# Thoracic UltrasONOgraphy Reporting: The TUONO Consensus

**DOI:** 10.3390/diagnostics13091535

**Published:** 2023-04-25

**Authors:** Italo Calamai, Massimiliano Greco, Marzia Savi, Gaia Vitiello, Elena Garbero, Rosario Spina, Luigi Pisani, Silvia Mongodi, Stefano Finazzi

**Affiliations:** 1Anesthesia and Intensive Care Unit AUsl Toscana Centro, Ospedale San Giuseppe, Viale Boccaccio 16/20, 50053 Empoli, Italy; italo.calamai@uslcentro.toscana.it (I.C.); rosario.spina@uslcentro.toscana.it (R.S.); 2Department of Biomedical Sciences, Humanitas University, 20072 Milan, Italy; marzia.savi@humanitas.it; 3Department of Anesthesiology and Intensive Care, IRCCS Humanitas Research Hospital, 20089 Milan, Italy; 4Laboratory of Clinical Data Science, Mario Negri Institute of Pharmacological Research IRCCS, 20156 Ranica, Italy; gaia.vitiello@gmail.com (G.V.); elena.garbero@marionegri.it (E.G.); 5Intensive Care Unit, Ospedale Generale Regionale Miulli, 70021 Acquaviva delle Fonti, Italy; luigipisani@gmail.com; 6Mahidol Oxford Tropical Research Unit, Bangkok 10400, Thailand; 7Anaesthesia and Intensive Care, San Matteo Hospital, 94403 Pavia, Italy; 8Laboratory of Clinical Epidemiology, Mario Negri Institute of Pharmacological Research IRCCS, 20156 Ranica, Italy

**Keywords:** point-of-care, lung ultrasound, report, modified Delphi, consensus

## Abstract

The widespread use of the lung ultrasound (LUS) has not been followed by the development of a comprehensive standardized tool for its reporting in the intensive care unit (ICU) which could be useful to promote consistency and reproducibility during clinical examination. This work aims to define the essential features to be included in a standardized reporting tool and provides a structured model form to fully express the diagnostic potential of LUS and facilitate intensivists in the use of a LUS in everyday clinical ICU examination. We conducted a modified Delphi process to build consensus on the items to be integrated in a standardized report form and on its structure. A committee of 19 critical care physicians from 19 participating ICUs in Italy was formed, including intensivists experienced in ultrasound from both teaching hospitals and referral hospitals, and internationally renowned experts on the LUS. The consensus for 31 statements out of 33 was reached at the third Delphi round. A structured model form was developed based on the approved statements. The development of a standardized model as a backbone to report a LUS may facilitate the guidelines’ application in clinical practice and increase inter-operator agreement. Further studies are needed to evaluate the effects of standardized reports in critically ill patients.

## 1. Introduction

The lung ultrasound (LUS) has emerged in the last decades as a highly sensitive and specific tool for the diagnosis of acute lung diseases both in Intensive Care Units (ICUs) and ordinary wards [1]. LUS clinical application has evolved from a rapid and focused point-of-care examination often limited to confirming or excluding an emergent condition to a test routinely performed in the ICU as part of the clinical examination [2]. In this perspective, a LUS is often carried out to monitor the evolution of pulmonary diseases, occasionally replacing a chest X-ray. 

Due to the large number of findings to be considered while performing a LUS, standardized LUS recording and reporting are pivotal to promote consistency and reproducibility during a clinical examination.

A standardized report could also improve the adherence to the published guidelines, as the type of report and its structure affect the quality of information reported and collected by physicians, and furthermore may increase inter-operator agreement [3]. As endorsed by Demi et al. in the recently published guidelines on the use of the LUS, the development of a standardized terminology and the implementation of LUS reporting with supportive images could improve clarity and patients’ safety [4].

To our knowledge, a comprehensive standardized tool to report LUS examination in the ICU has not been validated yet. On the contrary, our group previously investigated differences and similarities of LUS reports in a network of ICUs in Italy [5] and found significant variation in types of LUS reports and in the most reported items. 

Consequently, this work aims to define the essential features to be included in a standardized reporting tool and provides a structured model form to fully express the diagnostic potential of the LUS and facilitate intensivists in its use in daily clinical examination.

## 2. Materials and Methods

We conducted a modified Delphi process to build consensus on the items included in a standardized report form and on its structure. The modified Delphi technique is a well-known means to develop a consensus through subsequent rounds of anonymous evaluation of a list of statements, followed by the in-presence discussion of items [6]. A committee of 19 critical care physicians from 19 participating ICUs in Italy was formed, including intensivists experienced in ultrasound from both teaching hospitals and referral hospitals, and internationally renowned experts on the LUS [7]. Most physicians involved in the consensus had already participated in a study on the LUS in critically ill patients [8]. The committee was supported in technical and methodological aspects by the Gruppo Italiano per la Valutazione degli Interventi in Terapia Intensiva (GiViTI). The GiViTI has a two-decade experience in addressing quality measures and improvement in intensive care.

### 2.1. 1st Delphi Round

The 1st round used a structured anonymous multiple-choice questionnaire (MCQ) regarding the elements to be included in the LUS report (fully reported in Appendix A). It was conducted on 11 November 2019. Panelists were asked to express their judgment on 22 items on a 3-point Likert scale (“Agree”; “Neutral”; “Do not Agree”). Answers were collected using the Google forms platform. The list of items contained the possible elements and the structure proposed for a LUS report, and was derived from published literature [7,9,10,11] and the results of the TUONO 2 study. In the TUONO 2 study, we conducted a cross-sectional survey on LUS reporting and identified similarities and differences in the reports. The TUONO 2 study results were listed and completed with items from published literature and constituted the base for the 22 items presented in the first round. 

For each candidate item, panelists were also given the option to provide free-text comments to support their judgment, propose a change to an item, or add a different item.

The consensus threshold was set a priori at 80% of agreement. The questionnaire used in the 1st round is reported in the Appendix A.

### 2.2. 1st Virtual Meeting and 2nd Delphi Round

Results of the first round were presented at a virtual meeting held on 7 October 2020.

During the face-to-face meeting, anonymous results from the first meeting were discussed by participants. In items not reaching a consensus, participants were allowed to make proposals to introduce additional elements to the statements or propose improvement through rephrasing. 

Eight items were rephrased, and five were divided into two or more statements to improve quality. 

An updated questionnaire with 33 items was completed at the end of the virtual meeting. 

This 33-item questionnaire was anonymously administered to the panelists by email and constituted the 2nd Delphi round. 

As in the first round, panelists express their vote on the 3-item Likert’s scale (“Agree”; “Neutral”; “Do not Agree”). Answers were collected using the Google forms platform, and panelists were again able to express comments and suggestions in a free-text form.

### 2.3. 2nd Virtual Meeting and 3rd Delphi Round

Results from the second Delphi round were collected and discussed at a new virtual meeting held on 21 October 2021. A total of 28 statements reached a consensus during the second round. The other five items not reaching a consensus were discussed and rephrased to improve clarity and increase agreement.

At the end of the 2nd virtual meeting, the six modified statements were proposed for anonymous online voting through the Google survey platform. Four items reached a consensus (with an 80% threshold for consensus). In contrast, 2 items were excluded from the results.

All the experts and clinicians involved and contributing to the Consensus Conference are listed in Appendix A.

The process is displayed in Figure 1.

## 3. Results 

The panelists reached a consensus on 31 statements out of 33 statements. All the statements are presented in Table 1, including those rejected by the committee. 

Specifically, the following two statements (n° 26, n° 29) did not reach the threshold for consensus:n° 26: it is useful to report if the examination is performed in the emergency setting.n° 29: it is useful to report the type of probe.

## 4. Discussion 

### 4.1. Statements n° 1–5: Systematic Examination of Lung Zones and Localization of Findings

Several models using different thoracic LUS zones have been published throughout the last decade, often related to a single disease. Volpicelli investigated interstitial disease (pulmonary edema) and proposed a four-zone model investigating two anterior and two lateral zones per side, neglecting the posterior zones [10]. Similarly, Jambrik et al. proposed a 28-zone model to generate a score for extravascular lung water [12,13]. Even in this case, posterior zones were not considered. 

In critically ill patients, the interstitial syndrome is not the only encountered disease, and most pathological findings are localized in the posterior zones. Accordingly, Lichtenstein proposed a 3-zone model (anterior, lateral, and posterior) [1], while Via et al. proposed a 12-zone model including upper and lower, anterior, lateral, and posterior zones [14].

Compared to the 8-zone protocol and the 6-zone protocol, the use of a 12-zone protocol has the advantage of considering the posterior areas. Since the introduction of the Bedside Lung Ultrasound in emergency (BLUE) protocol [15], the 12-zone protocol has emerged as one of the most routinely used approaches in the critically ill. As shown by Tung-Chen et al., a 12-zone protocol is consistent with a higher degree of concordance with a CT scan, is more reproducible and facilitates inter-operator comparison [16].

We assume that the potential disadvantage of lengthening the examination aiming at a thorough investigation of the lungs may be surpassed by the net advantage of detecting earlier those conditions otherwise overlooked by antero-lateral scanning protocols. We endeavor a more precise localization of findings by exploiting anatomical landmarks (i.e., hemiclavear/axillary lines, number of the intercostal space) to evaluate any further evolution of the condition, especially for the location of small subpleural consolidations or lung point in case of small pneumothorax.

It is also preferable to report which zones have not been examined to make an exhaustive report: those zones that have been investigated but not described should be considered non-pathological.

### 4.2. Statements n° 6–8: Examination of the Pleura

Lung sliding and lung pulse are cornerstones for the exclusion of pneumothorax and should be accurately investigated in every situation where this disease is clinically suspected.

More frequently, when facing interstitial disease, describing pleural features as normal (regular) or pathological (irregular, fragmented, etc.) may be crucial in differential diagnosis between infective/flogistic or cardiac/renal etiology. These characteristics should always be assessed and reported if abnormal.

Moreover, the pleura has been extensively investigated during the COVID-19 pandemic. Pleural features have been promptly recognized as highly suggestive if not pathognomonic of COVID-19 pneumonia [17], ranging from the thickening of the pleural line to the presence of the typical *shred sign*, an irregular deep border corresponding to small subpleural consolidations separated by the surrounded aerated parenchyma [18].

### 4.3. Statements n° 9–12: LUS Signs

Unlike a conventional ultrasound, a LUS is primarily based on artifacts generated by lung-air interface. Each artifact has been associated with a specific pulmonary condition but not all the signs have the same value and application in daily practice [19]. Many LUS signs have been detected and proposed for vote. In statement n° 10, we underlined which signs could be considered redundant while reporting LUS compared to mainstay findings with a peculiar diagnostic value, especially in a critical patient who may rapidly decompensate.

Lung sliding, as an example, may effectively rule out pneumothorax with almost 100% negative predictive value [20] with low positive predictive value. Therefore, special consideration was made for the seashore sign and stratosphere sign: we consider them as redundant in confirming the absence or the presence of pneumothorax, which can be rapidly excluded by detecting sliding in the daily clinical practice.

We endorse the reporting of LUS signs rather than profiles [1].

### 4.4. Statements n° 13–14: Quantification of Findings

A quantitative approach for the assessment of the severity of the disease is recommended. While multiple approaches have been proposed for the quantification of the loss of aeration [11], a quantitative approach based on the computation of the LUS score [21,22] is advised only in selected cases, as further clarified in statement n° 20. In general, a qualitative description of the B-lines features should be preferred to other validated scores as the B-lines score [10] which is based on the count of B-lines and does not take into account the presence of consolidated areas; its most appropriate use is therefore to assess the presence of interstitial diseases in outpatient context.

### 4.5. Statements n° 15–18: Pleural Effusion

The LUS may detect pleural fluid from effusions larger than 20 mL, approaching the diagnostic performance of computed tomography. Historically, the LUS description of pleural effusion was semi-quantitative, described as minimal, mild, moderate, or severe. Another straightforward semi-quantitative approach is to report the extension of the effusion over the involved intercostal spaces, and its depth to calculate the volume of the effusion. Published formulas on effusion volume improve decision making on thoracic drainage. As reported by the statement n° 16, the position of the patient plays a key role not only in detecting the effusion but also in quantifying its volume (e.g., the Balik formula—patient supine with a mild (15 degrees) trunk elevation, the Eibenberger formula—patient supine). Robba et al. recently included the quantification of the effusion in a panel of basic US skills for intensivists [23].

### 4.6. Statement n° 19–20: Consolidations and LUS Score

We suggest describing the consolidation in a semi-quantitatively manner as a precise quantification could be challenging especially for inexperienced users. The description of bronchograms (static, dynamic) allows the differentiation between atelectasis and consolidation [1] and may improve the detection of ventilator-associated pneumonia.

The Lung Ultrasound score (LUSS) is a validated tool to assess regional and global lung aeration in ARDS [24] and enables the unification and standardization of different US reports; therefore, we endorse its application for acute respiratory failure, also to determine its evolution over time. Its use is suggested for the initial assessment and monitoring of interstitial syndromes as well as the early recognition of ventilation-associated pneumonias [25] While it may seem time consuming, in expert hands a complete exam only requires a few minutes and can be easily integrated into the daily assessment of ICU patients; however, as underlined by Robba et al. this is not a basic skill [23] and requires a dedicated training [26]. We do not recommend a routinary score evaluation as it requires a careful examination of each lung region in both the longitudinal plane and transversal plane [23,26]. In any case, if a quantitative approach is to be chosen for the critical patient, the LUS score, which is based on the percentage of the pleura occupied by artifacts (B-lines, subpleural consolidations), should be preferred to other scores because it is more complete. For instance, the B-lines score, which is based on the counting of the B-lines, does not take into account the presence of consolidated areas that cannot be neglected in the ICU patient [27].

### 4.7. Statements n° 21–23: Conclusions

After a systematic description of findings, we suggest reporting conclusions and recommendations in a dedicated free-text section. This free-text conclusions section minimizes the main limits of adopting a SR, such as non-reporting important information due to their exclusion from the standard model.

### 4.8. Statements n° 24–29: Context of Performing LUS

We endorse including in the report the elective/emergency setting to concisely indicate signs of deterioration and to allow monitoring of its clinical evolution. Moreover, we suggest specifying any difficulty encountered during the investigation.

Since the LUS is an artifact-based technique, technical factors might influence ultrasound findings, including the type of US device and the type of probe. Several studies aimed to determine the overlapping among results obtained using different probes [27]. This consensus did not reach an agreement on the choice of probe to be used; we focused on a report which balances the richness of details and pragmatism [28], hence any recommendation regarding the use of a specific probe is beyond the purpose of this work [27].

### 4.9. Statements n° 30–33: Ventilatory Setting and Therapeutic Consideration

It is noteworthy to consider how findings could change according to variations in ventilation; if neglected, this aspect could hamper a more precise quantification of aerated parenchyma under specific conditions and the assessment of the severity of illness and evolution over time.

The LUS allows the quantitative estimation of lung recruitment using the LUS score [29], and this can be considered for therapeutic and diagnostic decision [2,30,31]. We do not suggest including the indications as a report section, but to include them in the conclusion section.

### 4.10. Proposal for a Structured Model form for LUS

I.C., M.G, R.S., S.F., E.G., S.M. generated the TUONO reporting tool, a structured model form for reporting a LUS which aims to summarize the features that were agreed on from the Modified Delphi process. The model form was distributed to the panel of experts for editing and approval before the manuscript submission.

The aim of the model form was to facilitate the application of guidelines and the transmission of information between operators.

The structured form is reported in the Appendix A.

#### Limitations

To our knowledge, this is the first consensus on LUS reporting, designed to define the elements to be included in a LUS report and to develop a reproducible model to facilitate the guidelines’ application in clinical practice and to increase inter-operator agreement. The LUS is a point-of-care widely available tool, gaining a significant role as a first-line evaluation during the COVID-19 pandemic, specifically in contexts where a CT scan was not available or scarce due to the high surge of patients [32,33]. The downside of LUS diffusion is the variability in reporting and the lack of common ground to communicate, reproduce and compare results.

A limitation is the reduced number of ICUs involved in the project, which are part of a network of nonteaching centers and teaching hospitals in Europe (GiViITI network), dedicated to quality improvement in intensive care. Larger multicentric international studies are warranted to better assess the practices of reporting around the globe and compare the best options for LUS reporting. Another potential concern was related to the timeframe of this study; due to COVID-19 pandemic ebbs and flows, there was a 11-month-interval between round 1 and round 2. However, none of the experts withdrew their consensus.

The main advantages of Delphi method are the opportunity for a large panel of experts to participate in a consensus process, the ability to focus on ideas rather than on individual opinions, and to define priorities. The modified Delphi process has the further advantage of face-to-face rounds to share and discuss available information while avoiding experts’ bias during the successive anonymous surveys.

## 5. Conclusions

We presented the results of the first consensus on LUS reporting, designed to develop a reproducible model to facilitate the guidelines’ application in clinical practice and to increase inter-operator agreement. The panel proposed 31 items, to improve standardization in reporting and facilitate the LUS clinical practice, summarized in the proposal for a possible form. Further studies are needed to evaluate the effects of standardized reports in critically ill patients.

## Figures and Tables

**Figure 1 diagnostics-13-01535-f001:**
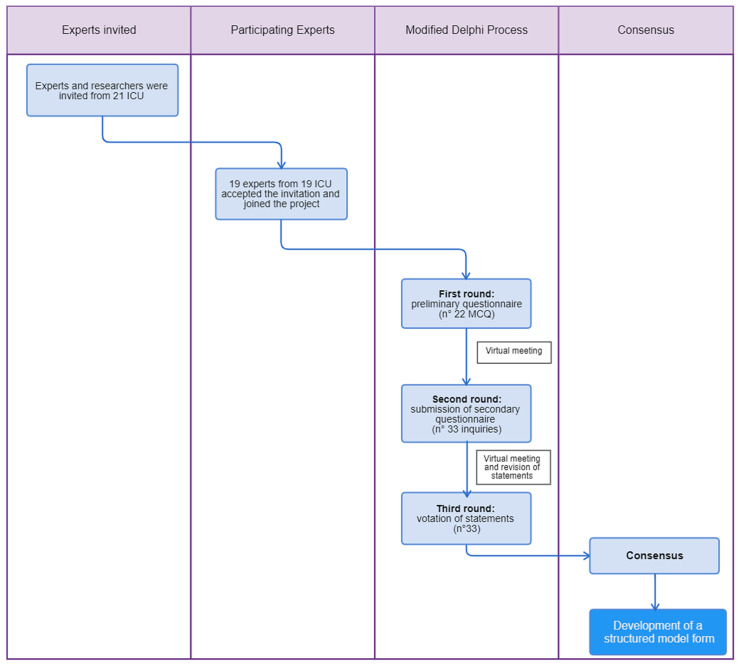
Modified Delphi process.

**Table 1 diagnostics-13-01535-t001:** Ultrasound findings approved by the consensus conference.

Experts’ Consensus Statements	Agree (%)	Disagree (%)	Neutral (%)
Systematic examination of lung zones			
It is preferable to use the 6-zone scanning protocol per each side to localize LUS findings to be reported in free text reports (FTRs) as well as standardized reports (SRs).	100	0	0
2. In case of specific findings, such as lung point or minimal subpleural consolidations, it could be useful to report a more precise localization in addition to the zone by indicating anatomical landmarks.	86.4	13.6	0
3. It is helpful to report which zones have not been examined.	92.9	7.1	0
4. If the report clearly states that all the zones have been examined, those not described should be considered normal without further explanation.	81.0	19	0
5. In a report, it should be clearly stated if any zone is normal, pathological, or not examined.	92.9	7.1	0
Examination of the pleura			
6. In a report, it should be stated if the pleura has been examined or not.	81.0	19	0
7. If the report clearly states that the pleura has been bilaterally examined and not described as pathological, it should be considered normal without further explanation.	100	0	0
8. In a report, it should be clearly stated if the pleura is normal, pathological, or not examined.	100	0	0
LUS signs			
9. B-lines, consolidations, effusions, lung point, lung sliding, static and dynamic bronchogram are keystone signs to be detected and reported in daily clinical practice.	100	0	0
10. A-lines, bat sign, curtain sign, lung pulse, quad sing, shred sign, sinusoid sign are signs that do not need to be reported in daily clinical practice.	100	0	0
11. Seashore and stratosphere signs are NOT fundamental signs to be detected and reported in daily clinical practice.	85.7	7.1	7.1
12. It is advisable to use US signs rather than US profiles.	92.9	7.1	0
13. Detected findings should be described upon their quantification or severity.	86.4	13.6	0
14. It is preferable to report a semi-quantitative and qualitative description of B-lines rather than their count.	92.9	7.1	0
Pleural effusion			
15. Pleural effusion should be described as mild, moderate, severe but it should be quantified by measuring its maximum thickness and reporting its extent.	100	0	0
16. The patient’s position during the quantification of pleural effusions should be reported.	100	0	0
17. The estimation of the volume, according to several formulas, could be a mainstay in the decision of draining a pleural effusion.	100	0	0
18. A qualitative description of a pleural effusion (hypoechoic, corpuscular, sepimented) must be solely in case the effusion is not free and anechoic.	100	0	0
Consolidations			
19. Consolidations must be described semi-quantitatively as small, moderate, extended. It is not necessary to measure their extent.	100	0	0
LUS score			
20. The report should provide a text box to calculate the LUS score, which is reserved for selected clinical cases (e.g., ARDS evolution monitoring).	92.9	7.1	0
Conclusions			
21. Each pulmonary zone should be described according to the findings detected, while interpretation and diagnosis should be reported in the conclusions section.	81.8	18.2	0
22. Clinical hypothesis should be reported in the conclusions section whenever the clinical case is clear or suggestive for a specific diagnosis.	100	0	0
23. The conclusions section should consist of a free text box to describe further elements not included in the SR.	100	0	0
Context of performing a LUS			
24. It is not mandatory to report the diagnosis or cause of the patient’s admission.	92.9	7.1	0
25. It is helpful to report if a LUS is performed as a screening examination or for monitoring a condition (follow-up)	92.9	7.1	0
26. It is useful to state if the exam is performed in emergency situation (REJECTED)	63.2	15.8	21.1
27. The position of the patient during the examination should be reported.	95.5	4.5	0
28. It is helpful to report technical difficulties and problems of execution of a LUS.	90.9	8.1	0
29. It is useful to report the type of probe (REJECTED)	70	30	0
Ventilatory settings			
30. It is helpful to report the type of ventilation of the patient.	91.0	9	0
31. Modification of findings according to variations in ventilation (e.g., recruitment maneuvers) should be reported.	95.5	4.5	0
Therapeutic considerations			
32. It is not necessary to report the diagnostic path.	100	0	0
33. It is not necessary to report the therapeutic path.	100	0	0

Rejected questions that did not reach consensus are displayed in red.

## Data Availability

Not applicable.

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
