# Peer review of "Thoracic UltrasONOgraphy Reporting: The TUONO Consensus"

_diagnostics, 2023, doi:10.3390/diagnostics13091535_

Round 1
Reviewer 1 Report
Major comments:
The authors emphasize that it is an international consensus (page 1, line 29; page 2, lines 67-68), but it is exclusively the consensus of Italian experts (including the name - TUONO - GROM). It is right to emphasize this in the abstract, methods and limitations.
Discussion; 4.1. – page 7, rows 128-150. The authors discuss the 6-zone protocol, the 8-zone protocol and the 12-protocol and conclude that the 12-zone protocol has advantages over the previous ones because it also includes posterior zones. Of course, a protocol with more zones, especially when posterior zones are included, is more precise, but such a protocol is time consuming and probably not always applicable in emergency conditions. This must be discussed in this part of the discussion.
Discussion, 4.6 – page 8-9, rows 200-209. It is difficult to agree with the author's statement that the routine assessment of LUSS is time-consuming, given that the proposed report with about thirty items is certainly much more time-consuming, and besides, there are simple software calculators for that. On the other hand, LUSS is a very important tool in unification and standardisation of different ultrasound reports. Also, simple LUS scores such as Modified Lung Ultrasound Sscore (Mongodi S, et al. Ultraschall Med 2017;38:530-537) do not require special training. This is very important, and please comment.
Minor comment:
This consensus is primarily aimed at patients in intensive care, so I think it should be highlighted in the introduction and conclusion. Namely, probably this form of report would not be entirely acceptable in emergency medicine.
Author Response
On behalf of my colleagues, I thank you for your comments and suggestions.
I will try to reply at the matters you highlighted:
- as you suggested, we further underlined that the consensus was based on a committee of Italian practitioners as well as internationally renowned Italian experts;
- we appreciated your suggestion. Regarding the 12-zone protocol, we acknowledge that this could seem more time-consuming, especially in the emergency setting, hence we reported that "the potential disadvantage of lengthening the examination aiming at a thorough investigation of the lungs may be surpassed by the net advantage of detecting earlier those conditions otherwise overlooked by antero-lateral scanning protocols” (line 145-147);
- regarding the paragraph dealing with LUSS, as you suggested, we underlined its importance as a validated tool to assess regional and global lung aeration in ARDS. We further expressed that, while its computation is not mandatory in every report and reserved to expert sonographers as reported by Robba et al. (PMID: 34787687), its use is strongly suggested for the initial assessment and monitoring of all the cases of lung acute respiratory failure, in particular interstitial syndromes as ARDS, as suggested by several studies (Bouhemad et al., PMID: 20851923, Mongodi et al., PMID: 28936711)
- we appreciated your suggestion and we tried to specify further that we directed this effort to improve the standardisation of US report specifically in the critically-ill patients
Best regards,
Massimiliano Greco
Reviewer 2 Report
Thoughtfully designed study that employed the Delphi protocol to identify the features of a LUNG POCUS report which are of highest relevance to the ICU setting. Notably, most of the 19 physicians involved had previously collaborated together on a multi-center survey of lung ultrasound use. As far as I can tell, all surveyed physicians worked in Italy.
Overall, I think this was a well-designed study and a well-written manuscript that has generated a sample lung POCUS reporting template. This template may be useful to standardize lung POCUS reporting in the ICU setting, although undoubtedly some variation will remain based on individual provider and institution preferences. For instance, at my institution, we use a 6-zone exam (3 views per hemithorax) that doesn’t map easily onto the template the authors have created and we have found our protocol suits our needs adequately. Our needs are limited primarily to urgent/emergent exams.
Please see any additional constructive feedback below.
Minor feedback:
Lines 141-143: “A 12-zone-protocol is also more 141 rapid to perform, reproducible, and facilitate inter-operator comparison and is now the 142 most routinely used approach in the critically ill.”
This is an interesting assertion but no references are provided to back it up. The authors should either (a) provide a reference backing up this contention that a 12-zone protocol is “now the most routinely used approach in the critically ill”
or
(b) reword this sentence to acknowledge that this 12-zone protocol is what the authors believe is commonly used across ICUs globally.
Lines 162-163: “…shred sign, an irregular pleural line corresponding to small 162 subpleural consolidations separated by the surrounded aerated parenchyma.”
The authors’ definition of “shred sign” is not consistent with how this term is normally used in the published lung POCUS literature. For example, Lichtenstein defines the shred sign as “the irregular border between consolidated and aerated lung” (PMID: 24401163) and NOT as an “irregular pleural line.” In fact, in my experience, it is possible to see a grossly normal-appearing pleural line in the presence of a subpleural consolidation.
Author Response
On behalf of my colleagues, I thank you for your comments and suggestions.
- Since the introduction of the Bedside Lung Ultrasound in emergency (BLUE) protocol (PMID: 18403664) the 12-zone protocol has emerged as one of the most routinely used approaches in the critically ill. As shown by Tung-Chen et al. (PMID: 34855187) a 12-zone protocol is consistent with a higher degree of concordance with CT scan, is more reproducible and facilitates inter-operator comparison. As suggested by the reviewer, we reported this protocol as one of the most used in ICU, not the most used one globally, and we tried to highlight the strong points of this approach.
- As suggested by the reviewer, we corrected the definition reported for the shred sign.
Best regards,
Massimiliano Greco